# Adaptive Cross-View Feature Mining for Jaw Lesions Detection and Recognition in CBCT Image

Wang Yue
School of Electronic and Information
Engineering
Beijing Jiaotong University
Beijing, China
22120143@bitu.edu.cn

Chen Houjin
School of Electronic and Information
Engineering
Beijing Jiaotong University
Beijing, China
hjchen@bjtu.edu.cn

Li Jupeng
School of Electronic and Information
Engineering
Beijing Jiaotong University
Beijing, China
lijupeng@bjtu.edu.cn

Hu Piaolin
School of Electronic and Information
Engineering
Beijing Jiaotong University
Beijing, China
22120067@bitu.edu.cn

Zhang Xinyue
Peking University School and Hospital
of Stomatology
Peking University
Beijing, China
1810303109@bjmu.edu.cn

Ma Ruohan
Peking University School and Hospital
of Stomatology
Peking University
Beijing, China
kqmrh@bjmu.edu.cn

Li Gang
Peking University School and Hospital
of Stomatology
Peking University
Beijing, China
kqgang@bjmu.edu.cn

*Abstract*—**Cone beam computed tomography (CBCT) plays a vital role in the jaw lesions clinical diagnosis. However, different types of jaw lesions exhibit similar appearances in CBCT slices, while existing computer-aided diagnostic models, neither 2D slices with lack of distinctive features or 3D volume with highly redundant information, resulting in limited performance. For better detection of jaw lesions, we proposed a novel cross-view feature mining detection network based on reinforcement learning to adaptively extract the most characteristic slices from multi-views. Specifically, for every transverse plane slice in the CBCT image, policy network is designed to extract these corresponding sagittal and coronal slices with the most critical features for lesion detection. And then these slices are encoded and fused into the recognition branch which enhanced the overall performance. In our experiments, the proposed network reached detection recall of 79.7%, precision of 89.2%, and high average precision (AP) of 0.84 with an intersection-over-union (IoU) of 0.5. Quantitative results show that the proposed network is more effective than existing advanced approaches in the clinical detection and recognition of jaw lesions.**

*Keywords—CBCT, jaw lesions detection, cross-view, feature mining, reinforcement learning*

## I. INTRODUCTION

Jaw lesions are the most common diseases in the human maxillofacial region, which present with a variety of different pathological characteristics [1, 2]. There are three common types of jaw lesions clinically: solid lesion, cystic lesion, and mixed lesion. Cone beam computed tomography (CBCT) is widely used in the clinical examination of jaw lesions, offering advantages such as high spatial resolution, low radiation dose, and imaging efficiency. However, clinical diagnosis of jaw lesions heavily relies on subjective evaluation of radiologists [3], leading to variability among practitioners [4], especially for lesions with unclear imaging features. Since the treatment options for these three types of jaw lesions are different, misdiagnosis of the lesions will lead to serious medical consequences. Therefore, computer-aided diagnosis has become a major research direction at present [5]. Yang et al. compared the performance of the YOLO network, oral and maxillofacial surgeons, and general practitioners in terms of precision, recall, accuracy, and F1 score, with the YOLO network outperforming human experts [6]. Yeshua et al. adopt Mask R-CNN for the lesions segmentation task (segmented as lesions or normal regions) in CBCT slices and combined the 2D results into 3D segmentation for subsequent lesion volume computation [7].

In the clinical diagnosis of jaw lesions, it is crucial not only to determine the presence of a lesion but also to determine the type to formulate an applicable treatment plan. Radiologists often rely on CBCT scans to confirm the cystic or solid nature of the jaw lesions. Subsequently, based on different density characteristics and in conjunction with pathological results, diagnosis results will be made. Accordingly, this paper classifies jaw lesions into these three categories: solid lesions, cystic lesions, and mixed lesions, also locates the lesion location. We introduced one novel method termed "adaptive cross-view feature mining" to dynamically extract the most distinctive feature slices from disparate views. Specifically, we employed one policy network based on reinforcement learning to identify and extract the most representative slices, i t mines features with the highest degree of uniqueness and relevance, thereby improving the performance of the entire network.

Our original contributions are listed below:

- We presented one feature mining policy network that leverages reinforcement learning to adaptively mine slices from multiple CBCT slices with the most distinctive features.

- We introduced one cross-view feature fusion strategy for 3D CBCT images to enhance recognition performance by integrating distinct features extracted from multi-views.

## II. RELATED WORKS

### A. Jaw Lesions Diagnosis

With the development of artificial intelligence (AI) and the increase in computational power, deep learning (DL) has shown great potential in the analysis of various medical imaging data [8] and has been applied in disease diagnosis, treatment assistance, prognosis prediction and other medical fields [9]. An increasing number of DL-based methods have been proposed for the classification, detection, and segmentation of medical dental images [10]. Studies have shown that the diagnostic accuracy of artificial intelligence in diagnosing oral diseases is comparable to or even surpasses that of professionals [11]. Lee et al. adopted the GoogLeNet Inception-v3 architecture to evaluate the classification performance of dentigerous cysts, keratocystic odontogenic tumors, and periapical cysts in CBCT and panoramic images. They enrolled 247 patients and used histopathologic results as the gold standard [12]. Literature [13] employed Inception v3 networks to categorize ameloblastomas and keratocystic odontogenic tumors in CBCT images. Jonas first categorized panoramic images into four classes (periapical cyst, periapical granuloma, other cysts, no lesions) using MobileNetv2, then employed YOLOv3 to detect lesions in images classified as periapical cysts or granulomas [14]. Ariji et al. utilized DetectNet to detect ameloblastomas, keratocystic odontogenic tumors, dentigerous cysts, radicular cysts, and simple bone cysts in panoramic images [15]. Kwon et al. modified YOLOv3 network to detect dentigerous cysts, keratocystic odontogenic tumors, periapical cysts, and ameloblastomas in 1282 panoramic radiographs, increasing the dataset size 12-fold via flipping, rotation, and intensity variation [16].

### B. Reinforcement Learning in Medical Image Analysis

In recent years, reinforcement learning has been widely used in the field of DL and has shown its superiority in various image processing tasks, especially in medical image detection applications [17]. Maicas et al. detected breast lesions from DCE-MRI images by implementing an attention mechanism through training an intelligent agent to search and focus on appropriate regions within the input volume, thereby effectively guiding the detection process [18]. Xu et al. takes computational challenge of breast cancer classification from histopathological images [19]. The selection network is trained using reinforcement learning, which outputs a soft decision about whether the cropped patch is necessary for classification. Alansary et al. evaluates multiple different reinforcement learning agents with training strategies for detecting anatomical landmarks in 3D images [20]. Yang et al. localized multiple uterine standard planes in 3D ultrasound simultaneously by one multi-agent DRL [21], which is equipped with one-shot neural architecture search (NAS) module.

## III. METHODS

### A. Overall Architecture

The structure of the detection of jaw lesions is depicted in the Fig. 1. Our network is mainly divided into two modules. The global feature extraction (GFE) module (composed of encoder and decoder), which is used to extract image features of one single two-dimensional CBCT slice. And the slices-feature mining and fusion (SMF) module, which is used to adaptively extract cross-view information for feature fusion.

Mirroring the clinical practice, axial-view image slice was selected as the principal-view slice $S_{pv}$, which serves as the foundation for subsequent detection and recognition tasks. And the sagittal- and coronal-views provide supplementary perspectives, as auxiliary-view slices $S_{av}$.

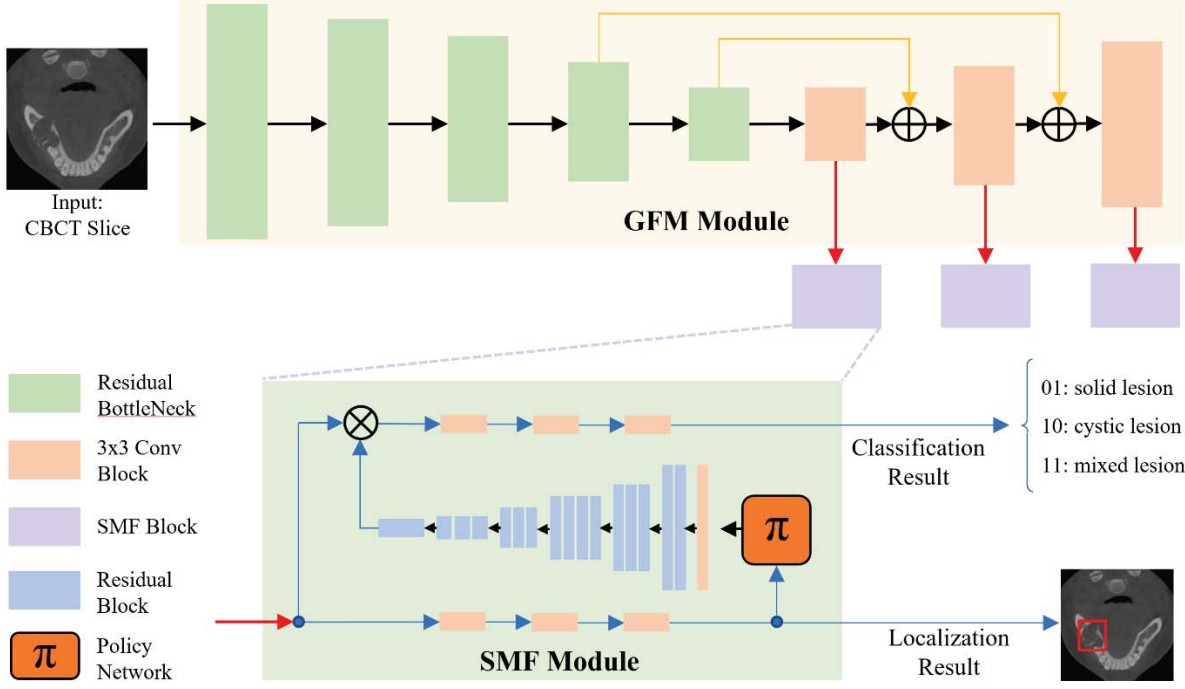

Fig. 1. The overall architecture of the proposed model.

## B. GFE network

The core process begins with the input of the $S_{pv}$ into convolutional neural network, where the initial feature extraction captures semantic nuances pertinent to the slice. The network architecture is fortified with residual blocks and integrated skip connections between the encoder and decoder, combining high-level semantic information with fine-grained spatial details. Following this, a feature pyramid framework is employed to generate feature maps across diverse scales, enhancing our ability to identify lesions of varying dimensions.

## C. SMF module

SMF block draws parallels to conventional dual-branch target detection heads, bifurcates its output between lesion classification and localization. Within this block, $S_{av}$ are adaptively extracted through the policy network, from which features are mined and fused into the jaw lesion recognition framework. This integration process is meticulously designed to augment the jaw lesions recognition network with the more comprehensive feature set derived from those multiple-view, thereby elevating the accuracy and robustness of the detection mechanism.

## D. Policy Network for Cross-View Slices Adaptively Extraction

In one CBCT image, slices from three viewing plane will intersect at one point in the 3D space, upon this, for each axial slice we select the sagittal and coronal slices that are most characteristic of the lesions, and the point of intersection of these three slices is what we call the best feature point $P_{bf}$. In other words, if we determine the $P_{bf}$, we obtain the slices from different views which present the most typical features for lesions recognition.

In SMF block, we first divide the bounding box from the localization branch into $n \times n$ regions evenly (here we take 3×3 as an example), and select the center point of each region as a candidate for $P_{bf}$. We extract coarse features from the axial slice, together with the $n \times n$ $P_{bf}$ candidates as inputs to the "π" network (Fig. 2). "π" is a policy network based on actor-critic (AC) algorithm [22]. The actor is responsible for selecting an action based on the current policy. Critic is responsible for evaluating how good the action taken by Actor is. The critic updates the value function based on the temporal difference (TD) error, which is the difference between the actual reward plus the discounted expected value of the next state and the value of the current state. The actor then updates its policy based on feedback from the critic. In this work, the actor will choose the $P_{bf}$ from the $n \times n$ candidate points: $\{P_1, P_2, \ldots, P_{n \times n}\}$. In specific, $P_{bf}$ is drawn from the distribution:

$$P_{bf} = \pi(g|e, c_n). \tag{1}$$

where $e$ denotes the coarse feature from the axial slice, $c_n$ denotes the $n \times n$ candidate points. After selecting one point $P_k$, π will receive a reward $r_k$ indicating whether this action is beneficial. Ideally, the reward is expected to measure the value of selecting $P_k$ in terms of lesion recognition. With this aim, we define $r_k$ as.

$$r_k = S_y(k) - S_y(0). \tag{2}$$

here $y$ refers to the label corresponding to the lesion, $S_y(k)$ refers to the softmax prediction score on $y$ (i.e., confidence on

the ground truth label), $S_y(0)$ refers to the score of $P_0$ (center of the bounding box) on $y$. That is, we define the reward value as the increase in confidence value of choosing $P_k$ as the $P_{bf}$ compared to choosing the center point of the bounding box as the $P_{bf}$.

After obtaining the $P_{bf}$, we determine the coordinates of the $P_{bf}$ in the axial slice, and together with the layer number of the axial slice in CBCT, we can determine the coordinates of $P_{bf}$ in 3D space: $(x_k, y_k, z_k)$. With $P_{bf}$, slices across three views are subsequently determined, the flow is shown in the Fig. 3.

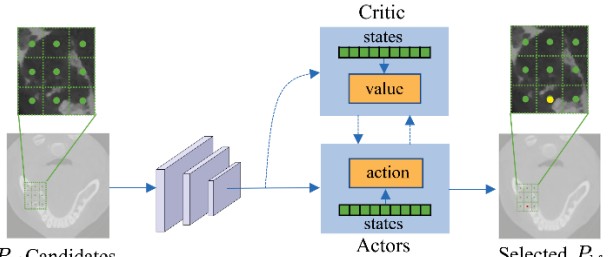

Fig. 2.   The architecture of the policy network π.

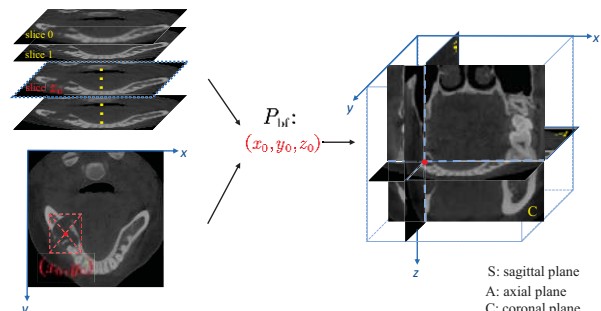

Fig. 3.   Process for determining cross-view slices.

## E. Cross-View Feature Fusion

One 2D image slice in the CBCT scan can be seen as one single channel image, thus, we can process the slices across the three views as a three-channel image in an RGB-like format. The cross-view slices encoding structure, as shown in the Fig.1, comprises several Residual Bottleneck units, each containing: one bottleneck layer to reduce the dimensionality of the input feature map, thereby reducing computational load; one 3×3 convolutional layer to extract features while maintaining the size of the feature map; and one expansion layer, one 1×1 convolutional layer, to increase the dimension of the feature map back to its pre-bottleneck size, helping to minimize information loss. After passing through several Residual Bottlenecks, the cross-view slices will undergo one convolutional layer to match the size of feature map of the main network, preparing for subsequent fusion.

Note that we used a convolutional block to make the encoded cross-view features have the same shape as the classification branch features of the detection network. We used multiplication fusion strategy, which multiplies these corresponding elements of each feature vectors. This above strategy is computationally less expensive, straightforward to implement, and can better capture nonlinear interactions between features, enhancing expressive power of the model.

With the above structure, the classification branch not only directly benefits from the optimization brought by the fusion

of cross-view information but also establishes a synergistic learning link with the location branch. This inter-branch enhancement addresses the issue of diminishing correlation between the classification and location branches.

## IV. EXPERIENT AND ANALYSIS

### A. Dataset

The dataset in this study comes from the School of Stomatology, Peking University and has received approval from the ethics review board. Pathological and radiological diagnoses were used as the gold standard for the data. The dataset was annotated and verified by a team comprising three radiologists with over three years of experience, one of them with over ten years of experience, and one senior radiologist. The inclusion types and criteria are as shown in Table I. For validation of our proposed framework, the dataset was divided into training set, validation set and test set at a ratio of 0.6:0.15:0.25. 2D slices in the axial planes were resized to 580×580 pixels in this process. Some example images are shown in the Fig. 4.

### B. Ethical Approval

This study was approved by the Biomedical Ethics Committee of Peking University School and Hospital of Stomatology (Approval Number: PKUSSIRB-202281152). Written informed consent was obtained from all participants prior to this study. All procedures were conducted in accordance with the Declaration of Helsinki.

TABLE I. INCLUSION/EXCLUSION CRITERIA USED IN THE STUDY

| Category | Inclusion Criteria | Exclusion Criteria |
|---|---|---|
| Clinical Records | Complete | Incomplete |
| Disease Type | ① Cystic lesions (42): periapical cyst, dentigerous cyst, odontogenic keratocyst, ameloblastoma. ② Solid lesions (22): odontoma. ③ Mixed lesions (13): ossifying fibroma. | Excluding cases with complex lesions, excluding cases with pathological diagnoses other than dentigerous cyst, odontogenic keratocyst, standard type ameloblastoma, odontoma, ossifying fibroma, as well as extra osseous/peripheral ameloblastoma, metastatic ameloblastoma. |
| Imaging Timing | Pre-operative CBCT images | Excluding patients with a history of previous surgery, malignant lesions, or recurrence; excluding postoperative X-rays |
| Image Quality | Images clearly showing the lesion areas of interest | ① Images with significant artifacts affecting the region of interest, including metal artifacts, motion artifacts, etc. ② Radiological images severely distorted, with artificial noise, blurring, and poor quality making them difficult to distinguish. |

### C. Training Details

We experimented with CUDA 11.1 on Ubuntu server 18.04 with 1 NVIDIA RTX 6000 Ada Generation GPU (48 GB). The designed framework is implemented using Python 3.7, and Pytorch. The axial slices sizes vary across different cases in the dataset. Thus, in the preprocessing stage, for axial slices smaller than 580 pixels, zero padding is applied to the edges to pad the images to 580×580. For axial slices larger than 580 pixels, up-sampling is used to unify the image size.

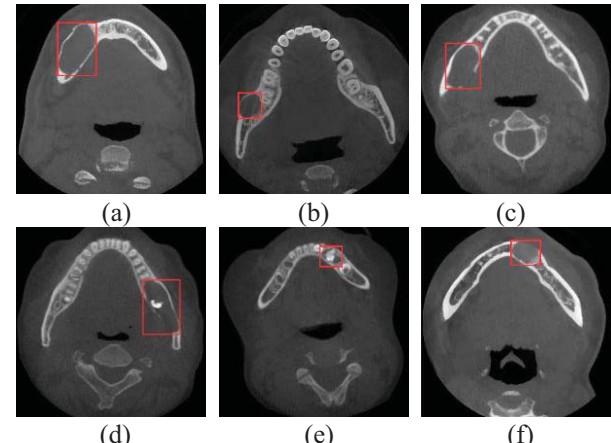

Fig. 4. Axial plane slices of CBCT images, with lesion areas marked in red boxes, (a)-(d) represent cystic lesions, (e) represents a solid lesion, and (f) is a mixed lesion.

### D. Visualization and Quantitative Results

The visualization results of our network's detections are shown in the Fig. 5, the first row shows the original axial slice to be detected, the second row presents the ground truth for detection, the third row displays the predicted results, and the fourth and fifth rows show the slices from the other two viewing planes mined by the policy network.

In our comparative experiments conducted on four networks (YOLOv8, Faster-RCNN [23], SSD [24], and DETR [25]), we considered one prediction to be correct if the intersection over union (IoU) between the predicted bounding box and the true bounding box exceeded 0.5, and incorrect if the IoU was less than 0.5. Under the condition of an IoU threshold set to 0.5, we obtained different categories of precision and recall values for five groups of networks as shown in the Fig. 6, where precision is denoted as $P$, and recall is denoted as $P$. The calculation formulas are as follows:

$$R = \frac{TP}{TP + FN}, \ P = \frac{TP}{TP + FP} \tag{3}$$

where $TP$ stands for True Positive, $FN$ for False Negative, and $FP$ for False Positive. It is observed that mixed lesions have very low recall value in the other four networks. This is due to the fact that mixed lesions exhibit different pathological characteristics at different slices, if we only identify the type of lesion from one single view, it is easy to misclassify them as other types of lesions. Our proposed network mines and integrates features from different views, compensating for the lack of distinctive features in single-view slice. This approach maintains the high precision for mixed lesions while achieving a recall rate of 0.701. Besides, the overall mean precision is 0.892, the recall is 0.797, which are all much higher than the other networks.

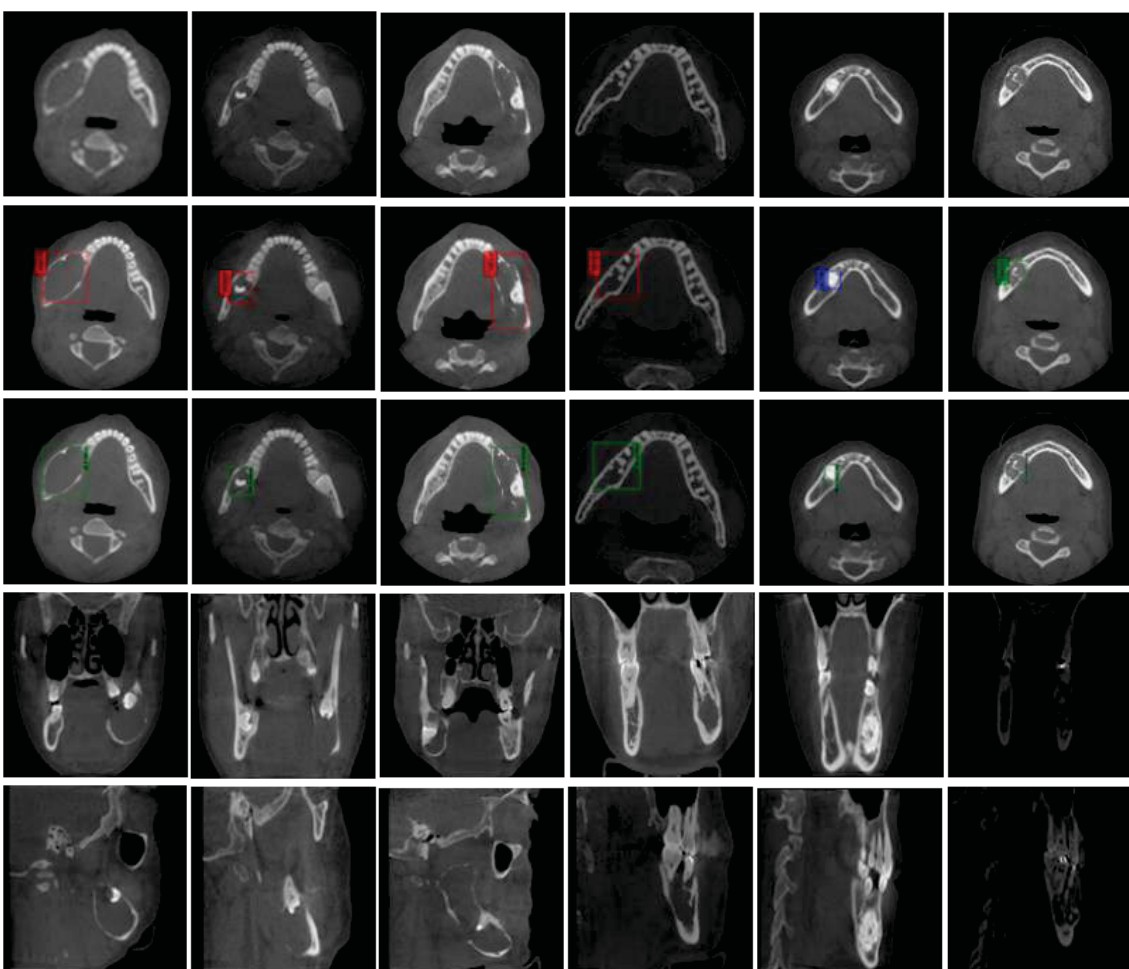

Fig. 5. Visualization results on detection of jaw lesions from different slices.

The mean average precision (mAP)50 value and mAP50-95 value, compared with other networks, are shown in Tables II and III. respectively. Here, mAP50 refers to the mean average precision at an IoU threshold of 0.5, while mAP50-95 represents the average precision across the range of IoU thresholds increasing from 0.5 to 0.95, with increments of 0.05 in our experiments. The network proposed in this paper achieved a mAP50 value of 0.843 and a mAP50-95 value of 0.695, both significantly surpassing the comparative networks.

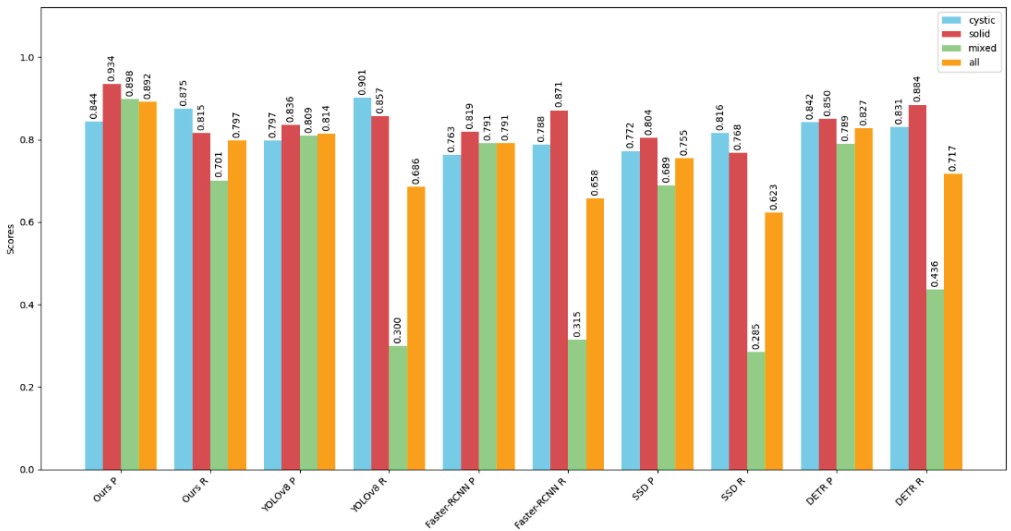

Fig. 6. The recall and precision results of different detection models.

Additionally, we conducted detection experiments on different $S_{pv}$: using the axial slice as the $S_{pv}$ with sagittal and coronal slices as $S_{av}$; employing the sagittal plane as the $S_{pv}$ with axial and coronal slices as $S_{av}$; and utilizing the coronal

slice as the $S_{pv}$ with sagittal and axial planes as $S_{av}$. The $R$, $P$ and mAP50 values from these experiments results are presented in Table IV.

TABLE II. mAP50 VALUE OF THREE TYPES OF LESIONS AND AVERAGE VALUE, THE RESULTS COME FROM DIFFERENT DETECTION MODELS：YOLOV8，FASTER-RCNN, SSD, DETR AND OUR PROPOSED MODEL, THE BEST RESULTS ARE IN BOLD.

| Models | cystic | solid | mixed | all |
|---|---|---|---|---|
| YOLOv8 | **0.828** | 0.891 | 0.521 | 0.747 |
| Fater-RCNN [23] | 0.767 | 0.894 | 0.514 | 0.725 |
| SSD [24] | 0.800 | 0.835 | 0.438 | 0.691 |
| DETR [25] | 0.780 | 0.902 | 0.574 | 0.752 |
| Ours | 0.807 | **0.944** | **0.778** | **0.843** |

TABLE III. mAP50-95 VALUE OF THREE TYPES OF LESIONS AND AVERAGE VALUE, THE RESULTS COME FROM DIFFERENT DETECTION MODELS：YOLOV8，FASTER-RCNN, SSD, DETR AND OUR PROPOSED MODEL, THE BEST RESULTS ARE IN BOLD.

| Models | cystic | solid | mixed | all |
|---|---|---|---|---|
| YOLOv8 | 0.684 | 0.612 | 0.388 | 0.561 |
| Fater-RCNN [23] | 0.683 | 0.599 | 0.332 | 0.538 |
| SSD [24] | 0.646 | 0.547 | 0.307 | 0.500 |
| DETR [25] | 0.654 | 0.622 | 0.395 | 0.557 |
| Ours | **0.734** | **0.730** | **0.621** | **0.695** |

TABLE IV. RESULTS FROM DIFFERENT $S_{pv}$, USING AXIAL, SAGITTAL AND CORONAL SLICE AS THE $S_{PV}$.

| Views | $P$ | $R$ | mAP50 |
|---|---|---|---|
| axial slice | 0.892 | 0.797 | 0.843 |
| sagittal slice | 0.875 | 0.763 | 0.826 |
| coronal slice | 0.859 | 0.791 | 0.832 |

*E. Ablation Study*

Here, we performed ablation studies to evaluate these enhancement. Baseline is a traditional dual-branch detection network without cross-view selection and fusion; baseline + cross-view feature fusion uses the center point of the bounding box as the $P_{bf}$ to determine the cross-view features and then fuse them into the classification branch; baseline + $P_{bf}$ adaptively selecting + cross-view feature fusion is the main method proposed in this paper. Table V shows the quantitative results of our ablation studies. With our proposed strategy, mAP50 value is improved by 0.121 compared to the baseline.

TABLE V. ABLATION STUDY RESULTS ON mAP50.

| Frameworks | cystic | solid | mixed | all |
|---|---|---|---|---|
| Baseline | 0.772 | 0.882 | 0.506 | 0.720 |
| Baseline + cross-view feature fusion | 0.803 | 0.943 | 0.741 | 0.829 |
| Baseline + $P_{bf}$ adaptively selecting + cross-view feature fusion | 0.806 | 0.946 | 0.772 | 0.841 |

*F. Discussion and Analysis*

In this study, we present a jaw lesions detection and recognition network that can adaptively select slices from different views of the CBCT image, and utilizes cross-view feature fusions to significantly enhance the performance of jaw lesions recognition. Meanwhile, the slices selection strategy based on the reinforcement learning is able to adaptively select the slices with the most typical features. The fusion of sagittal, coronal, and axial slices addresses the challenge of identifying lesions with similar radiographic appearance, as well as the difficulty of detecting lesions from one single 2D slice. Our results show that cross-view fusion based on adaptive selection strategy greatly improves the accuracy of detection and recognition, highlighting the potential of adaptive cross-view fusion technology in medical imaging.

One of the main findings of this study is that our network is able to compensate for the lack of distinctive features in individual slices, which is a common limitation in CBCT image analysis. By adaptively selecting and fusion features from multiple views, our network is able to provide a more comprehensive representation of jaw lesions, leading to more reliable diagnoses.

However, this study also acknowledges certain limitations. The first aspect is the weak interpretability of the adaptive slice selection. Although our experiments show that the diagnostic accuracy of the network can be improved by fusing adaptively selected cross-view slices, we cannot accurately understand the reasons why the network chooses these slices over the other slices. Interpretability of network decisions is one of the most important factors that enable computer-aided diagnosis to be used in the clinic. The second is that the present network uses supervised learning, which relies heavily on well-annotated medical data, whereas for jaw lesions, the gold standard data must be supported by pathology results; however, not all patients with CBCT images have a pathologic diagnosis. In other words, there is limited labeled data that can be used as fully supervised learning. Future work will focus on enhancing the interpretability of the network and further utilizing data without gold standard to improve the performance of the network to further improve diagnostic accuracy.

## V. CONCLUSION

We present one method for detection and classification of jaw lesions in CBCT images through adaptive cross-view feature mining and fusion. Our approach, utilizing reinforcement learning, effectively identifies the most informative slices from different views, enhancing the model's ability to distinguish between lesions with similar appearances. The experimental results demonstrate the superiority of our method over traditional techniques, offering significant improvements in detection and recognition accuracy. Despite its promising outcomes, the study highlights the need for further research to improve the interpretability of adaptive slice selection and reduce reliance on extensively annotated datasets. Future work will focus on addressing these challenges, aiming to advance the application of deep learning in medical images and contribute to more accurate and efficient diagnostics in clinical practice.

## ACKNOWLEDGMENT

This work was supported in part by the National Natural Science Foundation of China under Grant 62172029, the Beijing Natural Science Foundation under Grant L232029 and the Capital's Fund for Health Improvement and Research (CFH 2024-4-4107) from the Beijing Municipal Health Commission. (Corresponding author: Li Jupeng, E-mail: lijupeng@bjtu.edu.cn).

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
