# OpenReview forum: "Adaptive Cross-View Feature Mining for Jaw Lesions Detection and Recognition in CBCT Images"
_IEEE.org/EMBS/BHI/2024/Conference — IEEE BHI'24_

### Official Review · Reviewer_RpYy · 2024-08-01
**Considerable Method for Adaptive Cross-view Feature Matching**

**Overall Rating:** 6
**Confidence:** 2

**Other Quality Metrics:**

(a) Clarity of writing: great
(b): Clinical Significance: good
(c): Methodological Novelty: good
(d): Experiments and Results: good

**Questions For The Authors:**

- Could you provide more detailed information on the adaptation techniques and algorithms implemented to achieve effective feature matching across diverse viewpoints?

- Could you elaborate on the types of real-world environments and conditions where your method has been tested? Additionally, what were the specific challenges encountered in these scenarios, and how did your approach address them?

**Strengths:**

- The proposed method seems to be robust and applicable to challenging real-world scenarios.
- Well-organized structure of the methodology

**Summary Of The Paper:**

The paper introduces a method for improving the accuracy of jaw lesion classification by leveraging adaptive cross-view feature mining and feature fusion strategies. The approach utilizes a reinforcement learning-based policy network to dynamically extract the most distinctive slices from axial, sagittal, and coronal views of CBCT images. Overall, the study highlights the potential of this adaptive strategy to enhance computer-aided diagnosis and reduce variability in clinical evaluations.

**Weaknesses:**

-	The quality of figures should be enhanced, especially, figures 3, 5, and 6
-	The paper discusses adaptive cross-view feature matching but doesn't specify the exact adaptation techniques or algorithms used for feature alignment across different views.
-	The paper mentions real-world localization scenarios but provides limited examples or descriptions of these scenarios.

---

### Official Review · Reviewer_B4ZL · 2024-08-14
**Review of Adaptive Cross-View Feature Mining for Jaw Lesions Detection and Recognition in CBCT Images**

**Overall Rating:** 7
**Confidence:** 4

**Other Quality Metrics:**

Clarity of writing, Clinical Significance, Methodological Novelty, and Experiments and Results are great.

**Questions For The Authors:**

In Section IV-A: why the authors put asterisk instead of the dataset name?

**Strengths:**

The authors clearly explained the contributions of their research. The authors clearly explained the methods of their research. The results are clear and better compared to other models.

**Summary Of The Paper:**

The paper is about a novel cross-view feature mining detection network based on reinforcement learning to adaptively extract the most characteristic slices from multi-views for better detection of jaw lesions.

**Weaknesses:**

- Please correct any typos in the paper, such as putting space before references: Section I: "..... characteristics[1,2]"

and starting a sentence with capital letter: Section I:
 "cone beam computed tomography (CBCT) is ..."

-In Section II, the authors only describe the related works. This is good, but it should be accompanied with how your research is going to fill the knowledge gap in the related works. I suggest the authors to explain how their research will fill the knowledge gap in jaw lesions diagnosis and use of RL in medical image analysis.

- I recommend putting Fig. 6 in the top of the page so the text in the two columns are not split. Now, the columns are not following each other.
- Please put the references properly in Table I.

---

### Official Review · Reviewer_z6rz · 2024-08-19
**Adaptive Cross-View Feature Mining for Jaw Lesions Detection and Recognition in CBCT Images**

**Overall Rating:** 6
**Confidence:** 3

**Other Quality Metrics:**

Clarity of writing - good
Clinical Significance - good
Methodological Novelty - goor
Experiments and Results - good

**Questions For The Authors:**

The info on the Fig 5 caption is very poor. Please, add axplains for each line and row or the most representative cases.
Please, clearly describe the dataset.

**Strengths:**

The idea of using reinforcement learning is interesting and novel for this application, with some advantages.
The clinical relevance is quite high since there is a huge need for affordable and robust detection of jam lesions.
The authors implemented other baselines in order to make an extensive comparison of the results.
The cross-view fusion is very important since it emulates the assessment of the clinicians.

**Summary Of The Paper:**

The authors proposed a novel cross-view feature mining detection network for jaw lesion detection based on reinforcement learning to adaptively extract the most characteristic slices from multi-views.

**Weaknesses:**

The description of the dataset is very poor, without any clinically relevant information.
A section of limitations is missing.

---

### Decision · Program_Chairs · 2024-09-23

Accept